# Prenatal Exposure to Locally Emitted Air Pollutants Is Associated with Birth Weight: An Administrative Cohort Study from Southern Sweden

**DOI:** 10.3390/toxics10070366

**Published:** 2022-07-01

**Authors:** Festina Balidemaj, Erin Flanagan, Ebba Malmqvist, Ralf Rittner, Karin Källén, Daniel Oudin Åström, Anna Oudin

**Affiliations:** 1Division of Occupational and Environmental Medicine, Department of Laboratory Medicine, Lund University, 222 42 Lund, Sweden; festina.balidemaj@med.lu.se (F.B.); erin.flanagan@med.lu.se (E.F.); ebba.malmqvist@med.lu.se (E.M.); ralf.rittner@med.lu.se (R.R.); karin.kallen@med.lu.se (K.K.); daniel.oudin.astrom@umu.se (D.O.Å.); 2Sustainable Health, Department for Public Health and Clinical Medicine, Umeå University, 901 87 Umeå, Sweden

**Keywords:** ambient air pollution, fine particulate matter, local air pollution, source-specific exposure, traffic-related air pollution, residential burning, wood-smoke, low birth weight, birth weight

## Abstract

While prenatal exposure to ambient air pollution has been shown to be associated with reduced birth weight, there is substantial heterogeneity across studies, and few epidemiological studies have utilized source-specific exposure data. The aim of the present study was, therefore, to investigate the associations between local, source-specific exposure to fine particulate matter (PM_2.5_) during pregnancy and birth weight. An administrative cohort comprising 40,245 singleton births from 2000 to 2009 in Scania, Sweden, was combined with data on relevant covariates. Investigated sources of PM_2.5_ included all local sources together as well as tailpipe exhaust, vehicle wear-and-tear, and small-scale residential heating separately. The relationships between these exposures, represented as interquartile range (IQR) increases, and birth weight (continuous) and low birth weight (LBW; <2500 g) were analyzed in crude and adjusted models. Each local PM_2.5_ source investigated was associated with reduced birth weight; average decreases varied by source (12–34 g). Only small-scale residential heating was clearly associated with LBW (adjusted odds ratio: 1.14 (95% confidence interval: 1.04–1.26) per IQR increase). These results add to existing evidence that prenatal exposure to ambient air pollution disrupts fetal growth and suggest that PM_2.5_ from both vehicles and small-scale residential heating may reduce birth weight.

## 1. Introduction

Humanity’s burning of fossil fuels is not only a key contributor to climate change, but also produces air pollution. Air pollution, derived from both anthropogenic and natural sources, is the largest environmental cause of morbidity and mortality worldwide [1]. Ambient particulate matter (PM) with an aerodynamic diameter less than 2.5 µm (PM_2.5_) in particular accounts for a major part of the global burden of mortality and disease from air pollution. For instance, it was recently estimated that the fossil fuel component of PM_2.5_ causes 10.2 million premature deaths annually across the globe, or one in five deaths [2].

Regarding health effects attributable to air pollution exposure, pregnant women and their developing fetuses may be especially vulnerable. The current evidence of air pollution’s effect on birth weight and low birth weight is quite strong. A review evaluating 84 studies concluded that there were clear associations between maternal exposure to ambient PM and adverse effects on term birth weight and term low birth weight [3]. Considerable variation and inconsistencies between the included studies were noted, however. The authors, furthermore, concluded that to understand the relevant biological pathways, future research should focus on understanding which sources of PM are associated with term low birth weight [3]. A deeper understanding of whether all air pollution sources have the same health effects may not only help clarify underlying biological mechanisms but would also be valuable knowledge for policy decisions on air quality improvement.

Studies on source-specific exposure to PM and various health effects, including prenatal exposure on birth weight, are generally lacking due to the historical absence of such detailed exposure data. As source-specific exposure models have now been developed, the aim of the present study was to investigate source-specific ambient PM_2.5_ in association with birth weight and low birth weight in a low-exposure setting.

## 2. Materials and Methods

### 2.1. Study Setting and Study Population

The study was conducted in Scania (Skåne) county in southern Sweden, which had a total population of approximately 1.34 million in 2017. We defined an administrative cohort using data on 43,256 singleton births from 2000 to 2009 within the catchment area of hospitals in Malmö, Lund, and Trelleborg.

A detailed description of the cohort, Maternal Air Pollution in Southern Sweden (MAPSS), has been previously given elsewhere [4]. To create MAPSS, Perinatal Syd (PRS), a high-quality birth register with 98% coverage of all births in Scania [5], was used. PRS was linked to maternal socioeconomic and sociodemographic factors obtained from Statistics Sweden using each woman’s unique personal identification number. This data was then connected to individual exposure assessments of air pollution concentrations at each maternal residential address.

The study population was restricted to women who had given birth during the years 2000–2009 and for whom data on PM exposure and the child’s birth weight was available. We defined the remaining 40,245 singleton pregnancies as our final study population (Figure 1).

### 2.2. Exposure Assessment

A flat, two-dimensional Gaussian plume air dispersion model was used to model concentrations of ambient PM_2.5_. The dispersion model was developed using the software program ENVIMAN (OPSIS AB, Furulund, Sweden), a version of the American Meteorological Society/Environmental Protection Agency Regulatory Model (AERMOD) from the United States Environmental Protection Agency [6], which was adjusted to local circumstances. This model was used in combination with a detailed emission database (EDB). The EDB includes over 40,000 geocoded emission sources from road traffic; small-scale residential heating; railroads; shipping; aviation; industries and major energy and heat producers; nonroad vehicles; and emissions from Zealand, Denmark, which were gathered from various records, agencies, and authorities [7]. In-depth descriptions of the dispersion model and EDB have been previously published [7]. Using these air pollution data, annual mean concentrations of PM_2.5_ were modeled for the years 2000 and 2011 at a spatial resolution of 100 m by 100 m. To interpolate the monthly averages between the two modeled years, an atmospheric ventilation index with year- and month-specific meteorological parameters was utilized. This complex method is further detailed in a previous study [7]. With this, monthly means, approximately corresponding to the calendar months of the women’s pregnancies, for all local PM_2.5_ sources could be derived for each 100 m by 100 m grid cell. When validating the modeled PM_2.5_ values against measurement stations, the R^2^ varied between 0.44 and 0.86 [7].

The geographical coordinates of each woman’s residential address, obtained from Statistics Sweden, were linked to MAPSS and used to calculate individual exposure during pregnancy. We replaced pregnancies with missing monthly data with an estimated value based on the mother’s exposure in subsequent years using the expectation-maximization algorithm.

The focus of the present study was source-specific exposure to locally emitted PM_2.5_; therefore, regional background concentrations, from both naturally occurring air pollution sources and anthropogenic air pollution transported by wind into the area from long distances, were not considered. In Scania, regional background emissions typically comprise the majority of total PM. Thus, investigating local PM yields seemingly low exposure levels. The spatial contrast for regional background concentrations in this study area is low [7]. Because of this, effect estimates would still describe contrasts in local exposures even if regional concentrations were included. The interpretation of the estimates illustrates the effect of local contrasts rather than the effect of total PM, which has been more commonly studied in the current literature.

### 2.3. Exposure Variables

Four sources of locally emitted PM_2.5_ were investigated: all-source PM_2.5_, tailpipe exhaust, vehicle wear-and-tear, and small-scale residential heating (Figure 2). PM_2.5_ from wear-and-tear are generated from vehicles’ brakes and tires; in Scania, small-scale residential heating mainly consists of wood-burning. All other source contributions available from the emission database (railroads, shipping, aviation, industries and major energy and heat producers, nonroad vehicles, and emissions from Zealand, Denmark) were very small. Thus, these were not examined separately but are represented in the all-source PM_2.5_ category. The distributions of the exposure variables are seen in Appendix A.

### 2.4. Outcome Variables

The outcome variables in this study were birth weight (continuous; measured in grams) and low birth weight (LBW; dichotomous). For the latter, the World Health Organization’s (WHO) definition of LBW (<2500 g) was employed [8].

### 2.5. Covariates

The risk factors considered were maternal age (≤19, 20–34, ≥35), parity (1, 2, 3, ≥4), pre-pregnancy body mass index (BMI; <18.5, 18.5–24.9, 25–29.9, ≥30), smoking status at first antenatal visit (non-smoker, 1–9 cigarettes/day, ≥10 cigarettes/day, missing), fetal sex (male/female), birth year (2000–2009), and birth month from PRS. Additionally, we incorporated data from Statistics Sweden on individual socioeconomic status, namely, maternal education (≤9, 10–12, 13–16, and >16 years of education) and annual household disposable income (quartiles). Information on gestational length in days, estimated from ultrasound scans during pregnancy, was also available from PRS; extreme values (<175 days and >300 days) were coded as missing. Neighborhood-level socioeconomic status was included using a measure defined by Statistics Sweden as the proportion of inhabitants in the neighborhood with “low economic standard” based on income levels [9]. Whether the mother had preeclampsia (yes/no) during the pregnancy, defined as “severe or moderate” preeclampsia according to medical records, was also considered.

### 2.6. Statistical Methods

Due to high correlations between local PM_2.5_ sources (Appendix A), we utilized single-pollutant models. The main analyses assessed linear exposure trends using a linear interquartile range (IQR) increment increase in PM_2.5_ concentrations. The IQRs for each source were as follows: 0.99 µg/m^3^ for all-source PM_2.5_, 0.33 µg/m^3^ for small-scale heating, 0.12 µg/m^3^ for tailpipe exhaust, and 0.31 µg/m^3^ for vehicle wear-and-tear. To check the assumption of linearity, the associations were also investigated with the exposure variables transformed into tertiles. Linear regression was used when birth weight (continuous) served as the outcome of interest. When considering low birth weight (dichotomous), binary logistic regression was applied. Here, cases were defined as LBW babies and controls as non-LBW babies. Associations between locally produced PM_2.5_ and both outcomes were investigated using crude and adjusted models; the latter of which adjusted for the effects of maternal age, parity, pre-pregnancy BMI, smoking status, education as categorical variables, annual household disposable income (continuous variable), and sex of the child.

In separate sensitivity analyses, additional covariates were added to the adjusted model. The first is adjusted for birth year and birth month. Birth year was included to account for both annual variation in long-distance, in-transported air pollution and potential trends in diagnostic procedures. Birth month was utilized to adjust for seasonal variations in air pollution emissions from, for example, wood-burning and studded-tire use not captured by the exposure models, as well as other seasonally varying potential confounders, such as vitamin D [10]. The second included gestational days, as gestational age is strongly related to (low) birth weight. A third sensitivity analysis incorporated neighborhood-level socioeconomic status, which is becoming standard to adjust for in environmental health studies. In the fourth, preeclampsia was added to the adjusted model because this condition has been associated with both air pollution exposure and (low) birth weight [11]. In another sensitivity analysis, mixed models were used to take into account that some mothers had more than one child in the MAPSS and with dependent observations.

All statistical analyses were carried out using SPSS version 27 (IBM^®^, Armonk, NY, USA). Results for the linear and logistic regressions are reported as the absolute change in birth weight (grams) and odds ratios (OR) for low birth weight, respectively. A significance level of 0.05 was used in all analyses, yielding 95% confidence intervals (CIs).

### 2.7. Ethical Approval

The Lund University Ethical Committee approved this study prior to its realization (Permission Number: 2014/696).

## 3. Results

The relationships between the outcome variables and covariates are detailed in Table 1. Birth weight was positively correlated with annual household disposable income, and the proportion of children with low birth weight (<2500 g) increased as annual household disposable income decreased. The same was true for education: the average birth weight was lower for babies born to mothers with lower education than those born to mothers with higher education, and the lower the mother’s education, the larger the proportion of babies with low birth weight. In addition, low birth weight babies were more common among mothers younger than 30 and older than 35 compared to mothers aged 30 to 34, which was the most common age to give birth. The average birth weight decreased and the proportion of low birth weight babies increased with increased smoking intensity. Moreover, low maternal BMI (<18.5 kg/m^2^) was correlated with low birth weight, but a pattern was less clear for the remaining BMI categories. Regarding parity, trends were not consistent; however, some differences were seen, such as a higher probability of low birth weight if parity was 1 or ≥4.

Considering PM_2.5_ exposure in relation to birth weight and the covariates (Table 1), concentrations generally seemed higher for low birth weight babies and among lower socioeconomic status (both education and income) groups. Additionally, ambient PM_2.5_ exposure appeared to be greater among women with higher smoking intensity. For maternal age, parity, maternal BMI, and fetal sex, no strong trends regarding PM_2.5_ exposure were observed. No clear differences in PM_2.5_ exposure were seen for birth month either, but there were variations across birth years (data not shown).

The results demonstrated clear associations between each investigated local source of ambient PM_2.5_ and birth weight (Table 2). Birth weight decreased by an average of 34 g (95% CI: 26–43) in association with an IQR increase in all-source PM_2.5_ in the adjusted models. The corresponding reductions in birth weight for tailpipe exhaust, vehicle wear-and-tear, and small-scale residential heating were 33 g (95% CI: 25–42), 33 g (95% CI: 25–41), and 12 g (95% CI: 5–19), respectively.

For LBW, an association with all-source ambient PM_2.5_ in the crude model, with an OR of 1.16 (95% CI: 1.04–1.30), was observed (Table 3). In the adjusted model, the OR was attenuated and precision was decreased: 1.07 (95% CI: 0.93–1.23). Regarding the traffic-related sources, tailpipe exhaust and vehicle wear-and-tear were only weakly associated with LBW in both the crude and adjusted models. The only local source of ambient PM_2.5_ that remained statistically significant after adjusting for covariates was small-scale residential heating, with a crude OR of 1.26 (95% CI: 1.17–1.37) and an adjusted OR of 1.14 (95% CI: 1.04–1.26).

When tertiles were used instead of linear exposure variables, the confidence intervals for the estimates were generally very wide, but most of the estimates seemed to suggest that it was reasonable to assume linear associations (data not shown).

The sensitivity analyses incorporating birth year and birth month, gestational days, and neighborhood-level socioeconomic status had generally little influence on the effect estimates (Appendix A). However, there was a tendency for the OR for LBW associated with PM_2.5_ from small-scale residential heating to become slightly more attenuated when adjusting for birth year and birth month, and the precision decreased, with an OR of 1.09 (95% CI: 0.98–1.23). In the analysis where preeclampsia was added to the adjusted models, the average reduction in birth weight was 34 g (95% CI: 25–42) for each IQR increase in all-source PM_2.5_, which is comparable to the 34 g (95% CI: 26–43) decrease found when preeclampsia was not included. In line with this, the estimates of the LBW analyses were not affected by adding preeclampsia into the statistical models (data not shown). The final sensitivity analysis utilizing mixed models, furthermore, had only a marginal effect on the estimates (data not shown).

## 4. Discussion

### 4.1. Main Findings

In this administrative cohort study from southern Sweden, we observed statistically significant associations between exposure to ambient, locally emitted particulate air pollution during pregnancy and birth weight for PM_2.5_ stemming from traffic-related sources as well as from small-scale residential heating, which is mainly attributable to wood-burning in this setting. Furthermore, the risk of low birth weight (<2500 g) was seen to increase with increasing exposure to PM_2.5_ from small-scale residential heating.

The sensitivity analysis adjusting for birth year and birth month resulted in slightly attenuated ORs for the association between PM_2.5_ from small-scale residential heating and LBW. While this may imply some residual confounding due to temporal trends in the distribution of exposure or potentially confounding factors, such yearly or seasonal variations are an unlikely explanation of the findings. Still, this may partly contribute to the increased risk for LBW associated with PM_2.5_ from small-scale residential heating.

Interestingly, we observed statistically significant associations even though the local contribution to PM_2.5_ emissions is very low in our study setting (Scania, Sweden) and, therefore, typically accounts for quite a small proportion of the total PM_2.5_ concentrations. For example, the mean local concentration of all-source PM_2.5_ was 1.41 µg/m^3^, while the total concentration of PM_2.5_ in Malmö, a major city within the study area, averaged around 10 µg/m^3^ between 2016 and 2020 [12]. Moreover, these total concentrations are well below the European air quality directive of 25 µg/m^3^ annual mean, but still do not meet the WHO’s recently revised, stricter air quality guidelines of 5 µg/m^3^ annual mean [13].

### 4.2. Evidence from Current Literature

It is well established that maternal smoking during pregnancy reduces birth weight [14] and increases the risk for low birth weight (<2500 g) [14]. Even indirect exposure, such as through environmental tobacco smoke, has been reported to have similar effects [15]. Despite differences with respect to composition, content, and dose, there are evident parallels between exposure to tobacco smoke and exposure to air pollution. Both encompass the inhalation of toxic substances in the air, for instance. Most studies on prenatal exposure to ambient air pollution have typically used a measure of total air pollution for their exposure assessments. Few studies to date have examined distinct sources or components of air pollution. One such source-specific study carried out in Northern Italy demonstrated that preterm birth was associated with exposure to PM_2.5_ from traffic, organic and oil combustion, and secondary sulfates during pregnancy [16]. Another source apportionment study in California found that each interquartile range increase in PM_2.5_ exposure from secondary ammonium sulfate, secondary ammonium nitrate, and suspended oil during pregnancy increased the risk of term low birth weight [17]. Similarly, associations between vehicle exhaust particles and birth weight have been observed in a Swedish study from Stockholm with data on 187,000 pregnancies [18]. The authors found the linear decrease in birth weight to be 7.5 g (95% CI: 2.9–12) for an IQR increase (0.21 µg/m^3^) in exhaust particles during the entire pregnancy. The effect size was, thus, smaller than in the present study, where a 33 g (95% CI: 25–42) reduction in birth weight for each IQR increase (0.12 µg/m^3^) in PM_2.5_ from tailpipe exhaust was seen. These differences are interesting and difficult to explain, as the two studies are similar in many respects. For example, both are register-based, Swedish studies using similar exposure assessment methods. However, the Stockholm study adjusted tailpipe exhaust particles for ozone and paternal education.

It has furthermore been suggested that exposure to indoor air pollution from wood-fueled cook fires may affect birth weight through the same mechanisms as environmental tobacco smoke [19]. Indoor air pollution from wood, charcoal, crop residue, and animal dung has also been shown to have adverse effects on birth weight [20,21]. There is, furthermore, growing evidence indicating that ambient (outdoor) air pollution has negative effects on birth weight [22,23]. This association has also been supported by experimental studies [24]. Our findings, where we observed associations between locally emitted ambient PM_2.5_ from both vehicles and small-scale residential heating, and birth weight, therefore, add to these previous findings.

### 4.3. Biological Mechanisms

The underlying biological mechanisms driving the association between PM_2.5_ and low birth weight are not well known. Oxidative stress and chronic systemic inflammation triggered by air pollution exposure are thought to play a role in fetal development and growth as well as placental function, which, in turn, affect a baby’s birth weight [25]. In addition, the toxic effects of air pollutants may damage the placenta at an epigenetic level. For example, a study on human placentas using a mixed-effects model found that placental circadian pathway methylation was positively and significantly linked to third-trimester exposure to PM_2.5_ [26]. Many epidemiological studies, including ones performed in this study setting with the same study population, have shown that exposure to air pollution during pregnancy is a risk factor for preeclampsia [11,27], which is, itself, a risk factor for low birth weight [28]. Preeclampsia is, thus, a pathway through which air pollution exposure may contribute to low birth weight. In the present study, however, we observed no evidence of preeclampsia confounding the association between PM_2.5_ exposure during pregnancy and birth weight.

The results indicated an association between PM_2.5_ from small-scale residential heating (mainly wood-smoke in the present study), but not other investigated PM_2.5_ sources, and an increased risk of LBW. This may imply differences in chemical composition and toxicity among the air pollution sources. There is growing evidence of the detrimental health effects of wood-smoke. Human chamber studies, for example, have suggested wood-smoke exposure to be associated with increased arterial stiffness and decreased heart rate variability [29]. Additionally, a recent review identified seven publications involving controlled human exposures to wood-smoke investigating endpoints, such as oxidative stress, inflammation, lung function, cardiovascular function, and other self-reported symptoms [30]. Five of these include at least one statistically significant concentration–response (C–R), and, together, the evidence suggests a C–R increase in systemic oxidative stress following acute wood-smoke exposure [30]. However, another review cautions the evidence from controlled human exposures to wood-smoke as being premature, with the most consistency derived from studies assessing airway effects [31]. Given that the OR for LBW in association with PM_2.5_ from small-scale residential heating was not statistically significant in the present study, and even attenuated when adjusted for birth month and birth year, this particular result should be interpreted with caution. Moreover, such associations should be interpreted and investigated by other experts in the field, as additional research is needed to corroborate this finding.

### 4.4. Implications

The importance of normal birth weight has been established through numerous studies that underscore its contribution to physiological and psychological development throughout one’s life [32,33,34,35,36]. Low birth weight can contribute to increased neonatal morbidity and mortality, as well as the potential for worse health in adulthood [37]. Early effects on the immunologic, respiratory, gastrointestinal, and central nervous systems; delayed effects on motor, hearing, visual, cognitive, behavioral, and social-emotional function; and diverse effects on general well-being and development are quite well established [37]. The wider societal impact of low birth weight is also substantial. For example, the economic burden associated with preterm birth (which is highly correlated with low birth weight) in the USA was estimated to be at least USD 51,600 per infant (USD 26.2 billion in total) annually in 2005, approximately 65% of which could be attributed to health care costs [38]. With this, our results imply that prenatal exposure to air pollution from small-scale residential heating, specifically, likely has considerable individual, societal, and economic impacts in the study area.

Additionally, our results are in line with a growing body of research suggesting that locally emitted air pollution has more severe health effects than regional, background air pollution [39], which tends to comprise the majority of a pollutant’s total concentrations. Since the dose–response functions applied in health impact assessments are typically based on total air pollution concentrations, health gains from policy measures at the local level may often be underestimated. Evidence from a study in Hong Kong illustrated that policy actions aiming to reduce air pollution should lie at the local level, as local efforts were more beneficial compared to strategies employed at the regional level [40]. Combined, this evidence emphasizes that the reduction of local emission sources should be prioritized.

### 4.5. Future Research

The health effects of PM stemming from traffic, a major source of PM in most urban areas in high-income countries, has been extensively studied [41]. This is particularly true for on-road tailpipe exhaust. However, the ongoing electrification of the vehicle fleet observed over the past decade [42] is projected to continue in the future [43]. While this is positive for overall air quality in urban settings, the reduction of locally produced exhaust would increase the proportion of local PM derived from other sources, such as vehicle wear-and-tear and small-scale residential heating. Although associations were observed for PM_2.5_ from both tailpipe exhaust and vehicle wear-and-tear in the present study, the correlation between these two sources was very high (Appendix A); therefore, it is difficult to fully distinguish between them and separate their exposure effects. In future studies, it is important to better understand the potential health effects of these distinct sources.

Small-scale residential heating, also termed residential wood-burning, has been shown to be a significant source of ambient PM [44,45], and its incomplete combustion processes also create other toxic compounds as byproducts [46,47]. Despite its harmful elements, very little is known about the health effects of ambient wood-smoke emissions derived from indoor wood stoves and recreational fireplace use. This is partly due to a historical lack of high-quality exposure data. In recent years, however, source-specific models have been used by epidemiologists to study the long-term effects of PM from residential wood-burning, where associations with dementia and pediatric asthma have been suggested [48,49]. In addition to its demonstrated health effects, residential wood-burning has been identified as a potential challenge for air pollution reduction [50]. While most European Union (EU) countries have not regulated emissions from wood stoves, a 2022 EU Directive on eco designs for wood stoves has been proposed [51]. The WHO has also outlined policy suggestions regarding biomass and other solid fuel use [52]. Our results, which indicate an association between locally emitted PM_2.5_ from small-scale residential heating and low birth weight, highlight the importance of such policy implementations.

### 4.6. Methodological Considerations

A key strength of this study is the large size of the birth cohort (MAPSS) and the detailed maternal PM_2.5_ exposure estimates using high-resolution dispersion modeling in Scania, Sweden. Additional strengths include the avoidance of selection bias, access to a complete population, and the use of harmonized data. With this, the generalizability of our results should be high.

Despite its strengths, our study also has limitations. First, there is a measurement error in the modeled concentrations of air pollution. When these were validated against values from measurement stations, the R^2^ varied between 0.44 and 0.86. This proportion of variation could be considered acceptable; however, 56–14% of the variation remains unexplained by the exposure models in the present study. It would, furthermore, have been desirable to adjust for ozone and paternal education as potential confounders in our study, but no reliable data was available. A weakness in all studies on ambient air pollution that assess exposure at the residential address is that this outdoor concentration of air pollutants is a poor indicator of actual personal exposure; therefore, exposure misclassification can be present [53]. Even though this is the standard approach in air pollution epidemiology, we cannot completely rule out differential exposure misclassification, and subsequent bias, as an explanation for our findings. Additionally, we could not distinguish between the effects of different exposure windows, such as trimesters, in the present study, which would have been beneficial. This is partly due to issues with missing exposure data, especially in the first trimester. A lower proportion of missing data in the first trimester would have allowed for a more flexible exposure modeling method, such as distributed lag models. Finally, the available data on births included those only occurring from 2000 to 2009. Because a vehicle fleet’s composition and other trends in air pollution exposure can change over time, it would have been advantageous to analyze more recent data. This would have, for instance, enabled the investigation of whether the association between exposure to air pollution during pregnancy and birth weight has changed over time. Unfortunately, we did not have access to such data.

## 5. Conclusions

The present study adds to current evidence that ambient exposure to air pollution during pregnancy reduces babies’ birth weight and indicates that locally produced PM_2.5_ from both traffic-related sources and small-scale residential heating (mainly wood-burning) contribute to this association. Furthermore, our results identify an association between local PM_2.5_ from small-scale residential heating and low birth weight (<2500 g). This suggests that the ambient emissions from, and subsequent health impacts of, wood-burning should not be overlooked by policy makers when planning for climate change mitigation and air quality improvement at the local level. Overall, these findings strengthen the existing literature, demonstrating that adverse health effects of exposure to PM_2.5_ from local emission sources occur even in low-exposure areas.

## Figures and Tables

**Figure 1 toxics-10-00366-f001:**
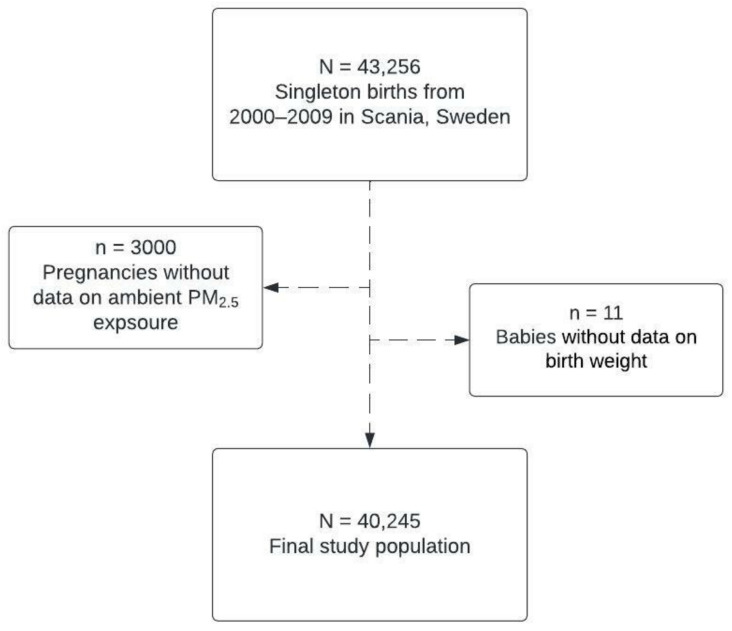
Flowchart of the study population by missing data on exposure (ambient PM_2.5_) and outcome (birth weight).

**Figure 2 toxics-10-00366-f002:**
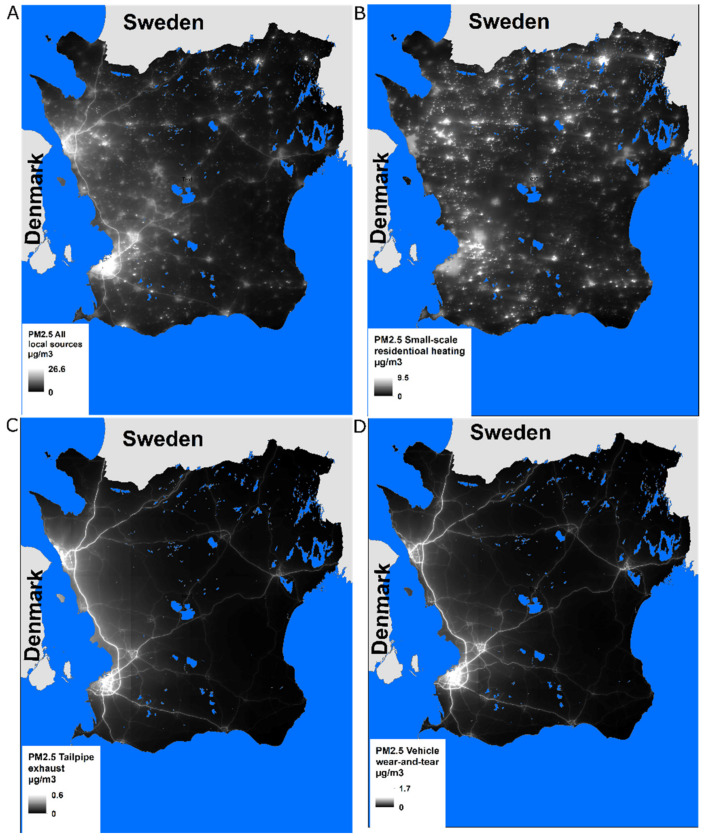
Annual mean PM_2.5_ concentrations from (**A**) all local sources, (**B**) small-scale residential heating, (**C**) tailpipe exhaust, and (**D**) vehicle wear-and-tear in µg/m^3^.

**Table 1 toxics-10-00366-t001:** Mean and standard deviation (SD) of local PM_2.5_ concentrations (µg/m^3^) during pregnancy, for birth weight, low birth weight (LBW; <2500 g), and different covariate categories, for children with and without LBW and totally. This table is based on the 40,245 study subjects for whom data on all-source PM_2.5_ was available.

	N *	Mean (SD)					(%)
		All-Source PM_2.5_	Small-Scale Residential Heating	Tailpipe Exhaust	Vehicle Wear-and-Tear	Birth Weight	LBW
Total	40,245	1.41 (0.64)	0.48 (0.25)	0.13 (0.08)	0.32 (0.21)	3564 (513)	1.8
LBW							
No	39,505	1.41 (0.64)	0.48 (0.25)	0.13 (0.08)	0.32 (0.21)	3589 (483)	–
Yes	740	1.47 (0.64)	0.54 (0.26)	0.14 (0.08)	0.32 (0.21)	2232 (257)	–
Annual household disposable income (quartiles)							
Lowest	9539	1.72 (0.63)	0.57 (0.28)	0.17 (0.08)	0.42 (0.22)	3467 (508)	2.4
Lower-middle	9910	1.52 (0.65)	0.53 (0.26)	0.14 (0.08)	0.35 (0.22)	3559 (521)	2.2
Upper-middle	10,733	1.25 (0.58)	0.44 (0.23)	0.11 (0.06)	0.25 (0.17)	3606 (513)	1.6
Highest	10,050	1.17 (0.52)	0.40 (0.20)	0.11 (0.06)	0.25 (0.17)	3617 (496)	1.2
Maternal education (years)							
≤9	4965	1.67 (0.63)	0.55 (0.27)	0.16 (0.08)	0.41 (0.22)	3477 (509)	2.6
10–12	16,854	1.35 (0.64)	0.47 (0.25)	0.13 (0.08)	0.30 (0.21)	3577 (525)	1.8
13–16	16,224	1.35 (0.60)	0.47 (0.24)	0.12 (0.07)	0.30 (0.20)	3592 (499)	1.6
>16	652	1.30 (0.51)	0.48 (0.22)	0.11 (0.06)	0.28 (0.16)	3567 (488)	1.4
Maternal age							
≤30	21,688	1.45 (0.64)	0.49 (0.25)	0.14 (0.08)	0.34 (0.22)	3536 (503)	1.8
31–34	10,904	1.34 (0.61)	0.47 (0.24)	0.13 (0.07)	0.30 (0.20)	3598 (518)	1.7
≥35	7653	1.37 (0.62)	0.48 (0.26)	0.13 (0.07)	0.31 (0.20)	3597 (530)	2.1
Maternal smoking status							
Non-smoker	34,193	1.40 (0.63)	0.48 (0.25)	0.13 (0.08)	0.32 (0.21)	3587 (500)	1.4
1–9 cig/day	2676	1.47 (0.65)	0.51 (0.25)	0.14 (0.08)	0.33 (0.21)	3426 (508)	2.3
≥10 cig/day	1091	1.51 (0.68)	0.53 (0.27)	0.14 (0.08)	0.34 (0.22)	3378 (522)	3.6
Maternal BMI							
<18.5	953	1.50 (0.64)	0.51 (0.25)	0.14 (0.08)	0.35 (0.22)	3316 (453)	2.9
18.5–<25	22,444	1.41 (0.63)	0.48 (0.25)	0.13 (0.08)	0.32 (0.21)	3530) 485)	1.6
25–<30	8731	1.41 (0.65)	0.48 (0.26)	0.13 (0.08)	0.32 (0.22)	3643 (513)	1.3
≥30	3732	1.42 (0.68)	0.48 (0.28)	0.14 (0.08)	0.33 (0.23)	3691 (550)	1.4
Parity							
First child	19,241	1.44 (0.64)	0.49 (0.25)	0.14 (0.08)	0.34 (0.22)	3491 (498)	2.4
Second child	13,396	1.33 (0.61)	0.47 (0.25)	0.12 (0.07)	0.29 (0.20)	3633 (514)	1.3
Third child	4973	1.37 (0.64)	0.48 (0.25)	0.13 (0.08)	0.31 (0.21)	3648 (521)	1.1
≥Fourth child	2635	1.62 (0.66)	0.55 (0.28)	0.16 (0.08)	0.39 (0.23)	3590 (530)	1.9

PM_2.5_: particulate matter with a diameter of <2.5 µm. Cig: cigarettes. BMI: body mass index. * Given for all-source PM_2.5_ (may vary slightly among PM_2.5_ sources).

**Table 2 toxics-10-00366-t002:** Decrease and (95% confidence intervals) of birth weight (grams) associated with an interquartile range (IQR) * increase in exposure concentrations of the investigated local PM_2.5_ sources during pregnancy.

	Crude	Adjusted ^†^
All-source PM_2.5_ ^‡^	59 (51–66), *p* < 0.001	34 (26–43), *p* < 0.001
Tailpipe exhaust	55 (47–63), *p* < 0.001	33 (25–42), *p* < 0.001
Vehicle wear-and-tear	51 (43–58), *p* < 0.001	33 (25–41), *p* < 0.001
Small-scale residential heating	33 (26–40), *p* < 0.001	12 (5–19), *p* < 0.001

* IQRs: all-source PM_2.5_ = 0.99 µg/m^3^, tailpipe exhaust = 0.12 µg/m^3^, vehicle wear-and-tear = 0.31 µg/m^3^, and small-scale residential heating = 0.33 µg/m^3^. ^†^ Adjusted for maternal education, annual household disposable income, parity, maternal BMI, maternal smoking at first antenatal visit. ^‡^ N = 40,245 and 33,853 for crude and adjusted models, respectively, and is given for all-source PM_2.5_ (may vary slightly among PM_2.5_ sources).

**Table 3 toxics-10-00366-t003:** Odds ratios and their (95% confidence intervals) of low birth weight (LBW; <2500 g) associated with an interquartile range (IQR) * increase in exposure concentrations of the investigated local PM_2.5_ sources during pregnancy.

	Crude	Adjusted ^†^
All-source PM_2.5_ ^‡^	1.16 (1.04–1.30), *p* = 0.007	1.07 (0.93–1.23), *p* > 0.300
Tailpipe exhaust	1.10 (0.99–1.24), *p* = 0.070	1.05 (0.91–1.21), *p* > 0.300
Vehicle wear-and-tear	1.01 (0.90–1.12), *p* > 0.300	0.97 (0.85–1.10), *p* > 0.300
Small-scale residential heating	1.26 (1.17–1.37), *p* < 0.001	1.14 (1.04–1.26), *p* = 0.007

* IQRs: all-source PM_2.5_ = 0.99 µg/m^3^, tailpipe exhaust = 0.12 µg/m^3^, vehicle wear-and-tear = 0.31 µg/m^3^, and small-scale residential heating = 0.33 µg/m^3^. ^†^ Adjusted for maternal education, annual household disposable income, parity, maternal BMI, maternal smoking at first antenatal visit. ^‡^ N = 40,245 and 33,853 in crude and adjusted models, respectively, and is given for all-source PM_2.5_ (may vary slightly among PM_2.5_ sources).

## Data Availability

The datasets generated during and/or analyzed during the current study are stored on a secure server and are not publicly available because they contain sensitive information (on health data, demographic characteristics, socioeconomic status) and, therefore, cannot be shared openly. However, they are available from Anna Oudin (anna.oudin@med.lu.se) on reasonable request.

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
