# Peer review of "Prenatal Exposure to Locally Emitted Air Pollutants Is Associated with Birth Weight: An Administrative Cohort Study from Southern Sweden"

_toxics, 2022, doi:10.3390/toxics10070366_

Round 1

Reviewer 1 Report

Birth weight and low birth weight are strongly related to gestational week, so why not consider gestational week? Or just use a full-term delivery.

Author Response

We agree with the Reviewer and have adjusted for gestational days in an additional sensitivity analysis. Information on gestational days has now been added to section 2.5 Covariates (Lines 131-133), the sensitivity analysis is named in section 2.5 Statistical Methods (Lines 159-160), and the results are described in section 3 Results (Lines 234-236 and included in Supplementary Tables 2 and 3).

Reviewer 2 Report

This administrative cohort study from Sweden shows that even at lower exposures(PM2.5<10ug/m3),Prenatal exposures,especially locally emitted particulate air pollution and Small-scale residential heating still adversely affect fetal birth weight, which provide evidence for air quality standards. There are a few questions that the author needs to answer.

1.    There have been many studies showing an inverse relationship between PM2.5 and low birth weight. The data in this study were from 2000-2009 in the Scania area, while reference 30 studied the data in the Stockholm area from 2003-2013. Why did the authors not analyze the most recent data to reflect the changes in air pollution in the past 10 years, and the research will become timely?

2.    The situation of the control group is not explained in the text, which should be stated in 2.1 study population.

3.    In 2.2, PM2.5 is simply divided into eight categories. Are references cited? {Rittner, 2020 #1165}? It is recommended to expand and explain the basis for such a classification.

4.    In 2.6, The IQRs for each source were as follows: 0.99 µg/m3 for all-source PM2.5, 0.33 µg/m3 for small-scale heating, 0.12 µg/m3 for tailpipe exhaust, and 0.31 µg/m3 for vehicle wear-and-tear.” How is the value of IQR determined? This data does not match the data provided by the boxplot of supplementary table2.

5.    Other studies on air pollutants mostly use distributed lag model (DLM) to analyze the cumulative effect or lag effect of outcome and air pollutants. It was not mentioned in your study whether it was possible to supplement the graphical associations of the four categories with low birth weight.

6.    The results of the sensitivity analysis on line 201 are not shown in the text and are suggested to be presented in the supplementary material.

7.    In the methods, sensitivity analyses included confounding factors such as birth year and birth month, neighbor-hood-level socio-economic status, and preeclampsia, whether based on empirical or data analysis, please specify.

8.    Section 2.6 does not state the criteria for the study to be statistically significant, for example: “To test statistical significance, 95% Confidence Intervals (95% CI) excluding the null were used.” Please add. In addition, P-values appear in table3, and P-values are not described in the Methods section.

9.    It is recommended to further explore why Small-scale residential heating has the greatest impact on low body weight among the 4 major pollutant sources, such as chemical composition? Is there any research to corroborate it?

Author Response

  1. There have been many studies showing an inverse relationship between PM2.5 and low birth weight. The data in this study were from 2000-2009 in the Scania area, while reference 30 studied the data in the Stockholm area from 2003-2013. Why did the authors not analyze the most recent data to reflect the changes in air pollution in the past 10 years, and the research will become timely?

Answer: We agree with the Reviewer that it would have been relevant to study these associations using more recent data. Unfortunately, we did not have access to newer data, which can be considered a limitation. We have, therefore, included a few sentences about this in section 4.6 Methodological Considerations (Lines 412-417).

  1. The situation of the control group is not explained in the text, which should be stated in 2.1 study population.

Answer: We have clarified in the manuscript how the controls were defined in section 2.6 Statistical Methods (Lines 147-148).

  1. In 2.2, PM2.5 is simply divided into eight categories. Are references cited? {Rittner, 2020 #1165}? It is recommended to expand and explain the basis for such a classification.

Answer: The sources of locally produced PM2.5 are not divided into categories or classifications after-the-fact; rather, data on these specific air pollution sources were gathered separately from various records, agencies and authorities and incorporated into the emission database. The text of the revised manuscript has been amended for clarification (section 2.2 Exposure Assessment, Lines 78-82). Additionally, {Rittner, 2020 #1165} is the proper reference, but it seems that there were some issues with the citation software. This has also been corrected.   

  1. In 2.6, “The IQRs for each source were as follows: 0.99 µg/m3 for all-source PM2.5, 0.33 µg/m3 for small-scale heating, 0.12 µg/m3 for tailpipe exhaust, and 0.31 µg/m3 for vehicle wear-and-tear.” How is the value of IQR determined? This data does not match the data provided by the boxplot of supplementary table2.

Answer: The IQRs was determined using SPSS, descriptive statistics. The data match the upper and lower levels of the boxes in the box-plots, (Supplementary Figures). We have double-checked and compared the IQRs to the box-plots but do not see any inconsistencies. Please let us know if anything more is unclear, and feel free to guide us further if, for instance, we have misunderstood the question.

  1. Other studies on air pollutants mostly use distributed lag model (DLM) to analyze the cumulative effect or lag effect of outcome and air pollutants. It was not mentioned in your study whether it was possible to supplement the graphical associations of the four categories with low birth weight.

Answer: In our experience, DLM can be problematic to use when there is a substantial amount of missing values in the exposure variables, which is the case in the present study (concerning monthly exposure). We, therefore, did not attempt a DLM or DLNM approach, and we were not able to analyze sensitive time windows such as trimesters separately in a satisfactory way. We fully agree with the Reviewer that DLM is an appealing approach in many settings, and that it would have been desirable to be able to investigate sensitive exposure windows during pregnancy. These limitations are elaborated upon in section 4.6 Methodological Considerations (Lines 407-412).

  1. The results of the sensitivity analysis on line 201 are not shown in the text and are suggested to be presented in the supplementary material.

Answer: We thank the Reviewer for noticing this and have added results from three of the sensitivity analyses to the Supplementary Materials.

  1. In the methods, sensitivity analyses included confounding factors such as birth year and birth month, neighbor-hood-level socio-economic status, and preeclampsia, whether based on empirical or data analysis, please specify.

Answer: We have now provided more detail on the sources of these variables in section 2.5 Covariates.

  1. Section 2.6 does not state the criteria for the study to be statistically significant, for example: “To test statistical significance, 95% Confidence Intervals (95% CI) excluding the null were used.” Please add. In addition, P-values appear in table3, and P-values are not described in the Methods section.

Answer: We thank the Reviewer for noticing this and have added a description of the criteria for statistical significance, including p-values, to the revised version of the manuscript (section 2.6 Statistical Methods, Lines 169-170).

  1. It is recommended to further explore why Small-scale residential heating has the greatest impact on low body weight among the 4 major pollutant sources, such as chemical composition? Is there any research to corroborate it?

Answer: Thank you for this recommendation. We have elaborated on possible explanations of this finding further in section 4.3 Biological Mechanisms (Lines 318-336). As very few source-apportionment studies exist to-date, particularly those investigating (low) birth weight as the health outcome of interest, no existing research could be identified to corroborate this result. We agree that more research on this association should be conducted and include this discussion within section 4.5 Future Research.

Reviewer 3 Report

This study strengthens existing reports that adverse health effects from exposure to local sources of PM 2.5 also occur in low-exposure areas. And it provides information for policymakers to use. It adds details to the current evidence that environmental exposure to air pollution during pregnancy reduces birth weight in newborns. Both transportation-related sources and small-scale residential heating (primarily wood combustion) contribute to this association.   As observation data, exposure data are well studied, and covariates, which affect LBW outcomes, are well-considered. The relationship between PM 2.5 and LBW has been discussed so far, and there is no major objection to the results. It is a well-designed study with adequate statistical and sensitivity analysis. On the other hand, it is unclear what was new in this study. The consideration of many factors is good, but the consistency of the results is difficult to read. I think it's better to say briefly what the authors want to say. In this study, SES was also used for Covariate. In particular, the effect of low LBW, as well as the effect of high PM 2.5, was reported. I think it will support the recently revised WHO guidelines and raise awareness of compliance.   I am not familiar with the environment of the study area, but is there a phenomenon that PM 2.5 increases or decreases depending on the season? Yearly variations are observed in many regions, but what about here? It may be common for exposure to vary by birth year.   The abstract seems to have a lot of sentences.   In some citations, the number format of the citation is not working. The code of the citation software is displayed.

Author Response

This study strengthens existing reports that adverse health effects from exposure to local sources of PM 2.5 also occur in low-exposure areas. And it provides information for policymakers to use. It adds details to the current evidence that environmental exposure to air pollution during pregnancy reduces birth weight in newborns. Both transportation-related sources and small-scale residential heating (primarily wood combustion) contribute to this association.   As observation data, exposure data are well studied, and covariates, which affect LBW outcomes, are well-considered. The relationship between PM 2.5 and LBW has been discussed so far, and there is no major objection to the results. It is a well-designed study with adequate statistical and sensitivity analysis. On the other hand, it is unclear what was new in this study. The consideration of many factors is good, but the consistency of the results is difficult to read. I think it's better to say briefly what the authors want to say. In this study, SES was also used for Covariate. In particular, the effect of low LBW, as well as the effect of high PM 2.5, was reported. I think it will support the recently revised WHO guidelines and raise awareness of compliance.  

I am not familiar with the environment of the study area, but is there a phenomenon that PM 2.5 increases or decreases depending on the season? Yearly variations are observed in many regions, but what about here? It may be common for exposure to vary by birth year.

Answer: Yes, it is true that pollutant levels may vary from year to year and depending on season. We, therefore, conducted a sensitivity analysis adjusting for both birth year and birth month to check that any such trends are not likely explanations for our findings (discussed in section 2.6 Statistical Methods, Lines 154-159).

The abstract seems to have a lot of sentences.  

Answer: Thank you for noting this. We have shortened the abstract to 200 words, as per the journal’s guidelines.

In some citations, the number format of the citation is not working. The code of the citation software is displayed

Answer: We apologize for this error and have amended the citations in the revised version of the manuscript.

Reviewer 4 Report

Thank you for having me as a reviewer for this paper. This study estimated the associations of locally emitted PM2.5 and its source-specific exposures during pregnancy with birth weight, using data from an administrative cohort in southern Sweden. The results showed that higher exposures to all-source PM2.5, and PM2.5 from tailpipe exhaust, vehicle wear-and-tear, and small-scale residential heating were associated with decreased birth weight; only PM2.5 from small-scale residential heating exposures were associated with an increased odds of low birth weight. The manuscript is generally well-written. Please see my comments below:

1) The dispersion model for PM2.5 predictions was originally developed in the U.S. and was adjusted to local circumstances. I am not able to get the information in English for citation #13. According to the paper titled “Particle concentrations, dispersion modelling and evaluation in southern Sweden” by Rittner et al. (2020), the R2 for the PM2.5 model ranges from 0.44-0.86; it seems the measurement error induced by the prediction models needs to be mentioned in the Methodological considerations.

2) I understand that the authors are interested in locally emitted PM2.5, so regional background concentrations were not included in the exposure assessments. If regional background concentrations had variations to some extent in the study area, should they be adjusted in the multivariate model?

3) How did the authors decide which covariates to be adjusted in the primary models and which covariates to be adjusted in the three extended models? What is the rationale of controlling for birth year and birth month? Even though neighborhood-level socio-economic status is not a strong confounder in these data, it has been considered an important confounder in environmental health studies – please consider including it in the primary models instead.

4) I agree that preeclampsia could be a mediator for the link from prenatal air pollution exposures to low birth weight. Since this analysis did not include a formal mediation analysis, I suggest re-writing the last sentence of “4.3. Biological mechanisms” – “In the present study, however, we observed no evidence of preeclampsia mediating the association between PM 2.5 exposure during pregnancy and birth weight” – to match the analytic approach in this study. 

5) The authors reported that they used exposure tertiles to check the assumption of linearity, but no results were shown for this analysis. There are a few “data not shown” in this study, including three sensitivity analyses and the sensitivity analysis using mixed models. Please considering add these results in the supplementary materials.

6) There are quite a lot of missing data in some of the covariates, e.g., parity 22928/40245 and maternal BMI 35860/40245. Did the authors implement any approach to handle the issue of missing data?

7) Some of the references are off. Please double check. 

8) Since the purpose of this study is not for causal inference, please consider avoiding the term “influence” in the title.  

Author Response

1) The dispersion model for PM2.5 predictions was originally developed in the U.S. and was adjusted to local circumstances. I am not able to get the information in English for citation #13. According to the paper titled “Particle concentrations, dispersion modelling and evaluation in southern Sweden” by Rittner et al. (2020), the R2 for the PM2.5 model ranges from 0.44-0.86; it seems the measurement error induced by the prediction models needs to be mentioned in the Methodological considerations.

Answer: We fully agree with the Reviewer that this is an important aspect of the prediction model. We have added this information to the methods (section 2.2 Exposure Assessment, Lines 89-91) and elaborated on its implications the discussion (section 4.6 Methodological Considerations, Lines 396-400).

2) I understand that the authors are interested in locally emitted PM2.5, so regional background concentrations were not included in the exposure assessments. If regional background concentrations had variations to some extent in the study area, should they be adjusted in the multivariate model?

Answer: The models were adjusted for birth year and birth month in sensitivity analyses for this reason (as well as for other reasons). This has been clarified and discussed in the revised manuscript within section 2.6 Statistical Methods (Lines 153-159), section 3. Results (Lines 234-239), and 4.1 Main Findings (Lines 254-259).

3) How did the authors decide which covariates to be adjusted in the primary models and which covariates to be adjusted in the three extended models? What is the rationale of controlling for birth year and birth month? Even though neighborhood-level socio-economic status is not a strong confounder in these data, it has been considered an important confounder in environmental health studies – please consider including it in the primary models instead.

Answer: We thank the Reviewer for their questions. The covariates in the adjusted models were chosen based on available data and a preliminary DAG. The rationale for controlling for birth year and birth month has been clarified and expanded upon in the revised manuscript (see section references and line numbers in the answer above). We have decided to keep neighbourhood-level socio-economic status within the sensitivity analysis; however, the results of this sensitivity analysis adjusting for neighborhood-level socio-economic status are now included in the Supplementary Material.

4) I agree that preeclampsia could be a mediator for the link from prenatal air pollution exposures to low birth weight. Since this analysis did not include a formal mediation analysis, I suggest re-writing the last sentence of “4.3. Biological mechanisms” – “In the present study, however, we observed no evidence of preeclampsia mediating the association between PM 2.5 exposure during pregnancy and birth weight” – to match the analytic approach in this study. 

Answer: We agree and have revised the manuscript accordingly.

5) The authors reported that they used exposure tertiles to check the assumption of linearity, but no results were shown for this analysis. There are a few “data not shown” in this study, including three sensitivity analyses and the sensitivity analysis using mixed models. Please considering add these results in the supplementary materials.

Answer: The confidence intervals for the tertiles were generally wide, and the results are not completely conclusive, but the estimates suggest linear associations for most analyses. We have clarified this in the manuscript (section 3. Results, Lines 231-233). We also agree that there were too many instances of “data not shown” and have, therefore, added results from sensitivity analyses to the Supplementary Materials.

6) There are quite a lot of missing data in some of the covariates, e.g., parity 22928/40245 and maternal BMI 35860/40245. Did the authors implement any approach to handle the issue of missing data?

Answer: We agree that the proportion of missing is quite high for some variables, as is shown in Table 1 as well as in the footnotes of Tables 2 and 3 (N=40,245 and 33,853 in the crude and adjusted models, respectively). Regarding parity, specifically, it seems there was a typo in Table 1 which incorrectly indicated a large number of missing. This has been rectified in the revised manuscript. To handle the issue of missing data, we considered using multiple imputation to replace missing values in the covariates. However, we ultimately decided not to because the overall percentage of missing in the adjusted models is 16% compared to the crude model, which we consider to be quite low.

7) Some of the references are off. Please double check. 

Answer: We thank the Reviewer for noting this and have double-checked the references.

8) Since the purpose of this study is not for causal inference, please consider avoiding the term “influence” in the title.  

Answer: We agree with the Reviewer and have revised the title of the manuscript.